# Lay attitudes and misconceptions and their implications for the control of brucellosis in an agro-pastoral community in Kilombero district, Tanzania

Caroline M. Mburu[1]*, Salome A. Bukachi[1], Kathrin H. Tokpa[2], Gilbert Fokou[2], Khamati Shilabukha[1], Mangi Ezekiel[3], Bassirou Bonfoh[2], Rudovick Kazwala[4], Katharina Kreppel[5]

1 Institute of Anthropology, Gender and African Studies, University of Nairobi, Nairobi, Kenya, 2 Centre Suisse de Recherches Scientifiques en Côte d'Ivoire, Abidjan, Cote d'Ivoire, 3 Muhimbili University of Health and Allied Sciences, Dar es Salaam, Tanzania, 4 Sokoine University of Agriculture, Morogoro, Tanzania, 5 Nelson Mandela African Institution for Science and Technology, Arusha, Tanzania

* mwihakicarol@gmail.com

**Data Availability Statement:** All relevant data are within the manuscript and its supporting information files.

## Abstract

Brucellosis is a priority zoonotic disease in Tanzania that causes ill-health in people and affects livestock productivity. Inadequate awareness and behavior risking transmission can impede control efforts. We conducted a cross-sectional survey of 333 livestock owners in three villages in the Kilombero district, Tanzania, to understand their awareness, knowledge and behavior associated with brucellosis. Six Focus Group Discussions (FGDs), two in each village, were conducted, as well as an additional FGD with male herders from one of the villages. Factors associated with knowledge on brucellosis, food consumption and animal husbandry behavior risking transmission of this disease, were identified using generalized linear models. Predictors for knowledge of brucellosis were being male and having a higher educational level, while age was positively associated with a higher level of knowledge. Faith and ethnicity were associated with the performance of practices risking transmission. Following traditional religion and belonging to the Wamaasai ethnicity significantly increased the odds of carrying out these practices. Qualitative analysis gave insight into risk practices and reasoning. Of the 333 respondents, 29% reported that they had experienced abortions in their herds, 14% witnessed retained placentas, and 8% had seen still-births in their cattle within the previous year. However, survey results also showed that only 7.2% of participants had heard about brucellosis as a disease in livestock. Of those who had heard about brucellosis in livestock, 91% associated abortions with it and 71% knew that humans can get infected through raw milk consumption. People overwhelmingly attributed symptoms and transmission of brucellosis in livestock to infection with trypanosomiasis and to supernatural reasons instead. In the community, consumption of raw milk was valued and handling of aborted material was not considered a risk for infection. This agro-pastoralist community holds on to long-held beliefs and practices and lacks understanding of the biomedical concept of brucellosis. Transmission routes and symptoms of brucellosis in humans and livestock are completely unknown. The disparity between risk perception and actual

**Funding:** The research for this paper was carried out within the framework of the DELTAS Africa Initiative [Afrique One-ASPIRE /DEL-15-008] (to CMM). Afrique One-ASPIRE is funded by a consortium of donor including the African Academy of Sciences (AAS) Alliance for Accelerating Excellence in Science in Africa (AESA), the New Partnership for Africa's Development Planning and Coordinating (NEPAD) Agency, the Wellcome Trust [107753/A/15/Z] and the UK government. The authors were funded by Afrique One ASPIRE to conduct this study. The funders had no role in study design, data collection and analysis, decision to publish, or preparation of the manuscript.

**Competing interests:** The authors have declared that no competing interests exist.

transmission risk related to animal handling and consumption of animal products presents a challenge for disease awareness communication. This study recommends focused community engagement and sensitization to address the limited awareness and misconceptions among agro-pastoralists.

## Author summary

Brucellosis is a zoonotic disease, causing abortion, retained placenta, infertility and reduced milk production in livestock and wildlife. In humans it leads to a febrile-like illness with headaches, joint pains, weight loss and arthritis. Approximately 500,000 people are diagnosed with brucellosis each year, mainly in sub-Saharan Africa. Risk factors for transmission are handling aborted material, residing with livestock and consuming raw milk and blood. Due to the cultural attachment that agro-pastoralists have with their livestock, they are most affected and it is a challenging disease to control. We conducted a study with agro-pastoralists in three villages, neighboring wildlife conservation areas, in south-central Tanzania. The majority of those interviewed did not know about brucellosis, its symptoms and risk factors for people and livestock. Symptoms associated with brucellosis infection in animals are recognized but are attributed to the disease trypanosomiasis and to supernatural reasons instead. Age, education and gender, as well as religious orientation and ethnicity were factors associated with knowledge and following risky practices. Behavior like consuming raw milk and blood, handling aborted material and co-habiting with livestock are considered an important part of life. We therefore suggest to policy makers, researchers and public health workers to engage with local communities and respond to their questions and concerns so that culturally and contextually relevant solutions are developed to control this disease.

## Introduction

Brucellosis is a serious bacterial disease, transmitted from livestock to humans through direct contact with animal birth or abortion materials or via the consumption of raw milk, meat or blood [1, 2]. As a neglected zoonotic disease, it is endemic in sub-Saharan Africa, Asia, parts of Europe and Latin America and is responsible for up to 500,000 human cases each year [3, 4]. It presents as a non-specific febrile illness in humans, commonly including body aches and headache, but can become chronic and cause more severe illnesses [5]. In livestock, brucellosis leads to serious economic production losses estimated at 6–10% of income per animal [6]. It also leads to low fertility and decreased milk production due to abortions causing lactating animals to produce up to 25% less milk [6]. The Brucella species, *B. melitensis*, which has the highest pathogenicity and infectiousness, and *B. abortus*, have been identified as the cause for most human cases [4]. Human behavior, such as handling livestock birth material and consuming raw animal products, has a fundamental role in the transmission of brucellosis to people [7]. Brucellosis in humans can also be acquired via direct contact with infected animals through skin abrasions or inhalation of infected aerosols [3, 4, 7].

Studies in sub-Saharan Africa observed that the highest incidence of brucellosis in both humans and livestock is often found in agro-pastoral systems due to the communities' attitudes and practices related to food consumption and animal handling [8, 9]. Several studies demonstrated the importance of assessing social and cultural understandings of risk [10, 11], as perceptions and attitudes were found to play a role in determining whether people take

preventive actions against brucellosis [10]. In different parts of Africa it was demonstrated, that there is poor and varying knowledge regarding the causes, symptoms and risk factors of brucellosis in both livestock and humans [12, 13].

In Tanzania, brucellosis prevention and control in humans and livestock has been prioritized, especially in agro-pastoral systems [14]. A high prevalence of infection ranging from 1–30% in livestock and 0.7–48.4% in humans has been reported from various regions [15–17]. This is partly due to entrenched practices related to the consumption of raw animal products and livestock handling, as well as due to difficulties in the implementation of control measures [16, 17]. There are numerous studies that have focused on the epidemiological aspects of brucellosis in Tanzania, but the resulting control efforts have not yet been successful. While brucellosis control strategies including test and slaughter and vaccination are working in developed countries, this is not the case in Sub-Saharan Africa [18]. It has been acknowledged that this is partly due to a lack of in-depth understanding of the cultural and contextual reasons that govern local attitudes and behaviors related to livestock handling and consumption of animal sourced foods [5, 19]. This understanding by affected communities and their cooperation is crucial, because vaccination of animals to control brucellosis has to be accompanied by other measures such as the proper disposal of aborted fetuses and placenta [18].

Over the past 20 years, agro-pastoralists have immigrated to the Kilombero district and settled at the fringes of existing villages [20]. There is a paucity of studies in the area on the awareness, attitudes and practices of local communities related to brucellosis; yet a holistic understanding is important, because disease control strategies need to have the support of local communities to be successful [6]. This study therefore explored the disparities between risk perception and practices performed risking transmission of the disease in an agro-pastoralist setting. The findings contribute insights into awareness and perception of brucellosis in this district. These insights are essential for the development of suitable control and sensitization strategies for brucellosis in this area.

## Methods

### Ethics statement

The National Institute for Medical Research in Tanzania (NIMR/HQ/ R.8a/Vol.1X/3102) approved the study. Before the study, the Morogoro regional administration authorities, the Kilombero District council and subsequently community leaders gave permission for the study and each respondent gave written informed consent and strict confidentiality was maintained in managing the data.

### Study area

This study was conducted in the three neighboring rural villages of Lungongole, Sagamaganga and Signal in the Kilombero district of the Morogoro region (Fig 1) between March and August 2019. Previously, the prevalence of brucellosis was shown to be 14.3% and 14.6% in cattle and humans respectively in the district [15]. The villages for this study were selected purposively to cover the isolated agro-pastoralist communities found at the fringes of the farming community. The district was previously inhabited by peasant farmers only, but agro-pastoralists migrated to this area in the last twenty years [20].

### Study population

The study population consisted of agro-pastoralists mainly from the Wasukuma, Wamaasai and Wamang'ati ethnic groups. The Wasukuma practice both farming and livestock keeping

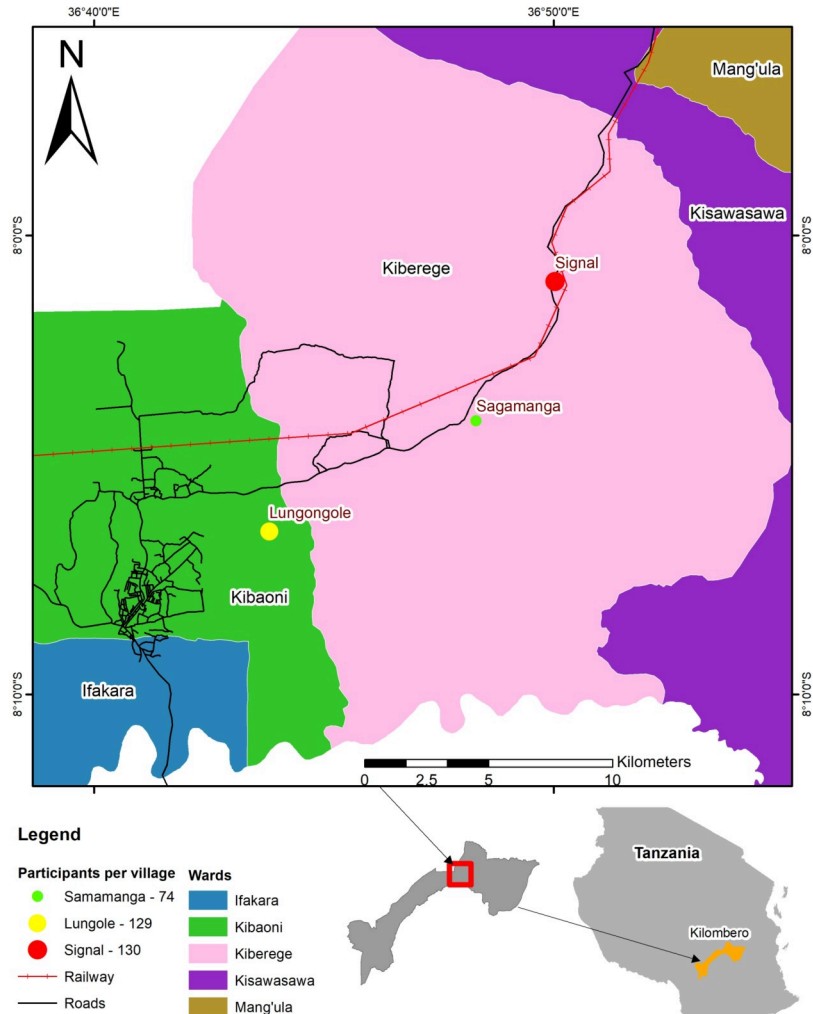

**Fig 1. Map of the Morogoro region indicating the three study villages where knowledge, attitudes and practices of agro-pastoralist communities were investigated.**

while the Wamaasai and Wamang'ati are mainly livestock keepers, farming only on a small scale if at all. These communities have continued to immigrate into this area in the last two decades in search of pasture and water for their livestock. While they keep sheep, goats and cattle, the latter are the most valued because of their greater economic value. These livestock keepers settled at the fringes of the existing farming villages from where they can easily access the communal grazing grounds located on the margins of the villages. The majority has no formal education, although this is changing and younger generations attend primary school. Most of these communities live a traditional lifestyle in close proximity to their livestock. They frequently self-medicate both livestock and people, using conventional and traditional medicines. Livestock is purchased from each other and from livestock markets spread throughout the district. Communal grazing grounds have been set aside where herding of cattle is done for most of the year. For about three months each year (March to May), the area is flooded and during that time the cattle are grazed within the homestead and in nearby open fields. This seasonal grazing pattern requires young herdsmen (12–30-year-olds) to live for up to nine months each year in makeshift structures in the communal grazing areas away from the homesteads.

## Study design, sampling and data collection

In this cross-sectional study quantitative and qualitative data were collected sequentially through structured interviews and focus group discussions (FGDs). First, quantitative data were collected on demographic variables, awareness of the major signs of brucellosis in live-stock and humans, knowledge of brucellosis transmission across the wildlife-livestock-human interface and risks of consumption of animal sourced foods, as well as animal handling practices and any observed brucellosis symptoms and perception of their causes (Supplementary Material 1). The project team (2 females and 1 male) visited all 176 agro-pastoralist households in the three villages together. In most participating households an adult male and an adult female were interviewed. Due to the unavailability of respondents in 9 households, a total of 333 interviews were conducted in 167 households. Consequently, a total of 129, 74 and 130 livestock keepers were interviewed in Lungongole, Sagamaganga and Signal villages respectively, of which 175 were male and 158 female. Only livestock owners and herders aged 18 years or older were included in the study. Pre-testing of the questionnaire was done with 20 livestock keepers in a neighboring village (Kikwawila) and those findings were not included in the final study. These respondents had a similar profile to the study participants as they were agro-pastoralists, immigrants to the area, and mainly from the Wasukuma ethnic group.

The same research team conducted follow-up interviews through 7 focus group discussions (4 male and 3 female groups) with 74 participants in total. In each village two FGDs were conducted (a total of 3 with men and 3 with women). One additional FGD was conducted with herdsmen in Sagamaganga village to obtain unique perspectives since the herders actually spent most of their young adult lives looking after livestock in isolated places. Each FGD had between 8–12 participants. Participants were purposively selected out of the respondents of the structured interviews with the help of each village livestock keepers' chair person. FGDs were conducted to gain deeper insights into zoonotic diseases including brucellosis, perceptions on risk factors for zoonoses, beliefs and practices related to raw milk consumption and animal husbandry practices.

Adult (≥18yrs) men and women were interviewed separately to avoid gender dominance in the interview. Interviews were held in a central location such as under a tree or in a primary school. Two trained research assistants, fluent in Kiswahili and Kisukuma, the main local language for most of the livestock keepers, were involved in the data collection to aid with comprehension. All the interviews were conducted in Kiswahili. During the focus group discussion, notes were taken by a research assistant and audio recorded.

## Data management and analysis

Quantitative data was analyzed using R software version 3.6.1 (Core-Team. R: A language and environment for statistical computing. *R Foundation for Statistical Computing*, *Vienna*, *Austria*, 2019). In order to quantify knowledge level, 29 categorical variables related to knowledge of brucellosis were individually scored on a binary scale of 0 and 1 as done by other researchers [21, 22]. Using the above scoring system, a total score was obtained for each respondent. The total score was then converted into a percentage score and the same procedure, now with a maximum score of 12, was used to assign a score for practices relating to brucellosis transmission pathways. For knowledge, the respondent who answered the most questions correctly scored the highest. For practices, the respondent reporting to perform the highest number of practices risking transmission scored the highest. The percentage scores for both knowledge and practices then formed the respective response variables for analysis.

Associations between socio-demographic variables and the level of knowledge of the disease were investigated with a zero-inflated negative binomial (ZINB) model, while

involvement in practices risking transmission was modelled using a generalized linear model (GLM). For the ZINB model the response variable was "knowledge score", while for the GLM the response variable was the proportion of risky practices undertaken of the total number of risky practices mentioned in the questionnaire. For both models' explanatory variables were "sex", "education", "age group", "religion" and "ethnicity" and for the GLM "knowledge" was added. For both models, explanatory variables "education" and "age-group" were treated as ordinal variables, while sex, ethnicity and religion were categorical variables. Age was transformed into age groups as ordinal numbers with ages 18–25 as group 1, 26–33 as group 2, 34–41 as group 3, 42–49 as group 4 and $\geq$50 as group 5. Education was changed into discreet ordinal numbers with 0 standing for no education, 1 for primary school education, 2 for secondary education and 3 for college/university education. Practice scores were transformed into proportions and a quasi-binomial distribution was used. Knowledge level was estimated as the sum of the 29 categorical variables, as described in more detail above, and knowledge scores were transformed into proportions. Since 93% (309) of participants answered that they had never heard of brucellosis and therefore scored zero in all questions on knowledge, we identified the data as over-dispersed due to the presence of excess zeroes and used a ZINB model [23]. This estimated the odds of socio-demographic variables being associated with having any knowledge of brucellosis and with the level of knowledge on brucellosis.

For all response variables of interest, model selection was based on backward elimination from an initial maximal model that included all predictors. Variables were tested using a likelihood ratio test and the variable with the highest p-value was removed at each iteration from the maximal model until only statistically significant terms remained. The overall model fit was assessed by plotting the coefficients and visually inspecting the residuals against the fitted values. Likelihood ratio tests were used to determine the influence of statistically significant variables. The packages 'pscl','lattice' and 'MASS' were used in R [24].

The FGDs recordings were transcribed into Swahili by one of the research assistants. Thereafter, the transcripts were translated into English and organized and prepared for analysis. The data were then entered into QSR NVIVO version 12.5.0 (NVivo qualitative data analysis software; QSR International Pty Ltd. Version 12, 2018). This social sciences software is used to order and organize qualitative data, to discover richer insights from qualitative and mixed methods research. In this study, the data were organized and grouped into codes that represented different themes regarding the knowledge, perceptions and practices related to brucellosis. This process was iterative, supported by existing literature, and the themes were modified and the relationships between them identified [25]. The emerging patterns were identified.

## Results

### Demographic characteristics

A total of 333 individuals participated in this study from 3 villages in the Kilombero district. A summary of the respondents' socio demographic characteristics is shown in Table 1. Of these 175 (53%) were male and 158 (47%) female with 129, 74 and 130 respondents from Lungongole, Sagamaganga and Signal villages respectively. Respondents were from the Wasukuma ethnic group (253, 76%) as well as Wamaasai (6.9%) and other agro-pastoral ethnic groups (17.1%). Most of the respondents were 34–41 years (30%) and 42–49 (24%) years of age with more than half (59%) having no formal education. Slightly more than half identified to practice traditional religion (56%) and 94% were married. Additionally, 74 (45 men and 29 women) of the initial respondents participated in the 7 focus group discussions.

**Table 1. Summary of the socio demographic characteristics of the respondents.**

| Demographic characteristics | | Structured interview respondents |
|---|---|---|
| **Sex** | Male | 53% |
| | Female | 47% |
| **Age** | 18–25 years | 7% |
| | 26–33 years | 19% |
| | 34–41 years | 30% |
| | 42–49 years | 24% |
| | >50 years | 20% |
| **Education** | None | 59% |
| | Primary | 39% |
| | Secondary | 2% |
| | Tertiary | - |
| **Religion** | Local religion | 56% |
| | Christian | 32% |
| | Muslim | 12% |
| **Ethnicity** | Wasukuma | 76% |
| | Wamaasai | 6.9% |
| | Others | 17.1% |
| **Marital status** | Married | 93.7% |
| | Separated | 0.9% |
| | Single | 2.4% |
| | Widowed | 3% |

## Awareness and knowledge of brucellosis in livestock and humans

Only a small minority (24, 7.2%) of the respondents had ever heard of brucellosis in livestock, while none had ever heard of the disease in humans. The respondents engaged in various practices that predisposed them to brucellosis. The variables for knowledge and practices are shown in Table 2 below.

Table 3 shows how many respondents were aware of brucellosis in livestock and/or animals and how many observed symptoms consistent with brucellosis in their herds in the previous year regardless of their awareness.

## Perceived sources of symptoms associated with brucellosis in livestock and humans

Over half of the respondents did not know the source of brucellosis related signs in livestock while some identified them as a result of trypanosomiasis as shown in Table 4 below.

According to the focus group participants, brucellosis related symptoms in livestock, especially abortions, were mainly attributed to trypanosomiasis. This was largely due to their observation that there were many tse tse flies in the grazing areas. These grazing areas were largely uninhabited and participants referred to them as *"mapori"* [the wild areas]. The following excerpts exemplify this: *"We know that when tsetse flies bite cattle then abortions occur and the cows do not produce milk."* (Women, FGD1, Kilombero) The same women added that they had never heard of brucellosis. Male participants from FGD3 made a similar statement: *"Tse tse flies are found in areas with wild animals and they cause abortions in cattle either after they bite the animals or when cattle eat grass that has tse tse flies eggs on it"*. Men FGD3, Kilombero

Other brucellosis related symptoms were also witnessed occasionally in the community including retained placenta, birth of weak or still born calves, reduced milk production and

**Table 2. Quantitative variables used to score knowledge and practices among the respondents.**

| Knowledge of brucellosis in livestock and humans | Number of participants responding (N = 333) |
|---|---|
| Awareness of brucellosis in livestock? | 24 |
| **Major signs of brucellosis in livestock? (N = 24)** | |
| Abortions | 22 |
| Reduced milk production | 17 |
| Birth of weak calves | 17 |
| Still births | 13 |
| Not conceiving/siring | 9 |
| Weight loss | 10 |
| Lameness | 6 |
| Swollen joints (hygromas) | 1 |
| Other signs attributable to brucellosis (premature calving) | 1 |
| Awareness of brucellosis transmission from wildlife to livestock | 21 |
| Awareness of brucellosis transmission from livestock to humans | 17 |
| **Brucellosis in humans (N = 333)** | |
| Awareness of brucellosis in humans | 0 |
| **Major signs of brucellosis in humans? (N = 0)** | |
| Fever | 0 |
| Chills | 0 |
| Headache | 0 |
| Joint pains | 0 |
| Abdominal pains | 0 |
| General malaise | 0 |
| Diarrhea | 0 |
| Vomiting | 0 |
| Sweating | 0 |
| Weight loss | 0 |
| **Transmission of brucellosis from livestock to humans? (N = 17)** | |
| Raw milk consumption | 17 |
| Raw blood consumption | 14 |
| Raw meat consumption | 15 |
| Direct contact with birth fluids | 17 |
| Inhalation of infected aerosols | 15 |
| Other signs (Physical contact with animal dung) | 2 |
| **Practices risking transmission with brucellosis** | **Number of participants responding (N = 333)** |
| Milking sick cattle | 100 |
| Consuming milk from sick cattle | 98 |
| Consuming raw milk | 263 |
| Slaughtering sick cattle | 177 |
| Slaughtering and skinning dead animals | 223 |
| Consuming meat from dead animals | 258 |
| Consuming meat from animals that died of natural causes | 251 |
| Consuming raw blood from animals | 34 |
| Consuming raw meat | 27 |
| Residing with livestock in the same house | 11 |
| Assisting in parturition with bare hands | 300 |
| Grazing livestock in areas with wild animals | 141 |

**Table 3. The awareness and experience of symptoms related to brucellosis among agro- pastoralists in the Kilombero district, Tanzania.**

| | Awareness | | |
|---|---|---|---|
| | Question | Total number answering "yes" (N = 333) | Frequency (%) |
| Awareness | Have you ever heard of brucellosis in livestock? | 24 | 7.2 |
| | Have you ever heard of brucellosis in humans? | 0 | 0 |
| Experience of brucellosis symptoms in livestock | Did you witness any cases of abortions in your livestock in the past one year? | 95 | 29 |
| | Did you witness any cases of retained placenta in your livestock in the past one year? | 45 | 14 |
| | Did you witness any cases of still births in your livestock in the past one year? | 28 | 8 |
| | Did you witness any cases of infertility in your livestock in the past one year? | 9 | 3 |

infertility. The agro-pastoralists assumed that these incidences were a result of unknown disease, a difficult birth process, body size and supernatural forces. These signs were also accepted as part of life and associated to similar incidences in humans especially in the cases of infertility and still births. Women said for instance: *"Still births happen because the animal was sick or in labor for a long time and also God had planned for that to happen just like it happens in human beings"*. (Women FGD 2, Kilombero). As a result of this acceptance, such livestock were retained in the herd, especially if other benefits were attributed to the animal, such as body size, calmness or a potentially good market price. The men mentioned that infertile cows were used as "draught animals" and kept for feasts:

*"We also keep them and slaughter them during important occasions at home because they often are very big in size"*. Men FGD 2, Kilombero

**Table 4. Answers to open questions on the perceived sources of common signs of brucellosis among agro-pastoralists in the Kilombero district, Tanzania.**

| Symptom in livestock | Perceived causes | Frequency (%) N = 333 |
|---|---|---|
| Abortions | Do not know | 56 |
| | Trypanosomiasis | 40 |
| | Unidentified disease | 3 |
| | Supernatural causes | 1 |
| Retained placenta | Do not know | 64 |
| | Trypanosomiasis | 16 |
| | Calving complications | 9 |
| | Unidentified disease | 7 |
| | Supernatural causes | 2 |
| | Brucellosis | 2 |
| Still births | Don't know | 50 |
| | Trypanosomiasis | 21 |
| | Unidentified disease | 21 |
| | Calving complications | 8 |
| Not conceiving/siring | Do not know | 78 |
| | Trypanosomiasis | 11 |
| | Supernatural causes | 11 |

Although participants, in all 7 FGDs did not know the causes of retained placenta, the remedies they used were perceived to be effective and thus this was perceived as a minor problem.

*"We wrap the protruding placenta around a stick and gently pull it over a few hours and eventually it comes out".*

Men FGD3, Kilombero

*"We have a local herb that we feed to the cows or salty water and the placenta comes off. It is never a big problem. Once we use that medicine the placenta comes out even after a number of days".*

Women FGD3, Kilombero

A major explanation for infertility in all the 7 FGDs was that it occurred as a result of increased body fat in an animal which impeded conception. The perceived duty of care for that animal and the perspective that the animal was still valuable as a traction animal or for slaughter prevents it from being removed from the herd. This is demonstrated through the following excerpts:

*"God gave me that animal to look after and he would be displeased if I sold it"*

Women FGD2, Kilombero

*"The cow was attractive to look at because it was very large and fat and would eventually be sold at a good price in the market or slaughtered in the home during an important ceremony because of its size".*

Men FGD2, Kilombero

The survey results show that more than three quarters (78%) of the respondents observed that they or a close family member had had a fever in the previous three months. In all the 7 FGDs participants did not know about brucellosis in humans but they knew about febrile illnesses, especially malaria, which was commonly experienced, followed by typhoid and urinary tract infections (UTIs). This is demonstrated through the excerpt below:

*"We have never heard of brucellosis and in the health facilities they don't mention it either. But when ill we are told we are either suffering from malaria, UTI or typhoid".*

Women FGD3, Kilombero

Other signs associated with a febrile illness were headaches (85%), abdominal pain (69%), vomiting (67%), joint pains (66%) and chills (63%). Focus group discussants noted that the major signs of a febrile illness were chills, fever and general malaise which were the most commonly mentioned (6/7FGDs) as well as headaches and stomach pain (4/7 FGDs). In this traditional agro-pastoralist community, the task of looking after livestock and managing livestock diseases belongs to men.

## Socio demographic characteristics associated with knowledge on brucellosis

The details of the respondents' knowledge related to brucellosis are shown in Table 2. The ZINB model provides two types of results (Table 5). The zero-inflation part shows the odds of

**Table 5. Predictors for knowledge and knowledge level of brucellosis by agro- pastoralists in Tanzania in a) the null model and b) the final model.**

| a) | Probability to have no knowledge, odds ratio (OR) | 95%CI | P-value | Level of knowledge, odds ratio (OR) | 95% CI | P-value |
|---|---|---|---|---|---|---|
| *Intercept* | 1142 | 64.8–20111 | <0.0001 | 13.5 | 3.73–47 | 0.0001 |
| sex (female) | 1 | - | - | 1 | - | - |
| sex (male) | 0.056 | 0.007–4.2 | 0.005 | 1.1 | 0.42–2.9 | 0.83 |
| education | 0.23 | 0.01–5.74 | 0.002 | 1.31 | 0.82–2.11 | 0.25 |
| age group | 0.69 | 0.46–1.05 | 0.09 | 1.3 | 1.11–1.53 | 0.001 |
| Religion (Traditional) | 1 | - | - | 1 | - | - |
| Religion (Christianity) | 1.93 | 0.62–6.04 | 0.25 | 0.63 | 0.32–1.21 | 0.17 |
| Religion (Islam) | 0.96 | 0.21–4.36 | 0.95 | 0.85 | 0.5–1.47 | 0.58 |
| Ethnicity (Wamaasai) | 1 | | - | 1 | - | - |
| Ethnicity (Other) | 0.76 | 0.65–1.02 | 0.92 | 1.5 | 0.98–1.7 | 0.96 |
| Ethnicity (Wasukuma) | 0.79 | 0.2–2.8 | 0.69 | 0.66 | 0.4–1.09 | 0.1 |
| b) | Probability to have no knowledge, odds ratio (OR) | 95%CI | P-value | Level of knowledge, odds ratio (OR) | 95% CI | P-value |
| Intercept | 915 | 74–1121 | <0.0001 | 12.143 | 3.44–42.8 | 0.0001 |
| sex (female) | 1 | - | - | 1 | - | - |
| sex (male) | 0.056 | 0.007–0.43 | 0.005 | 1.074 | 0.36–3.15 | 0.9 |
| education | 0.286 | 0.12–0.64 | 0.002 | 1.192 | 0.84–1.68 | 0.31 |
| age group | 0.71 | 0.47–1.06 | 0.09 | 1.228 | 1.05–1.44 | 0.01 |

having no knowledge are significantly decreased for males (OR = 0.05; 95% CI: 0.007–0.43) and by level of education (OR = 0.28; 95% CI: 0.12–0.64). Being older decreases the odds of having no knowledge of brucellosis (OR = 0.71; 95% CI: 0.47–1.06), but not significantly. The count part of the ZINB model addresses the odds of having better knowledge among knowledgeable group members. Age significantly increases the odds to know more by 22% (95% CI: 1.05–1.44). Being male increases the odds to have more knowledge by 7% (95% CI: 0.36–3.15), but this result is not significant. A better level of education increases the odds to know more by 19% (95% CI: 0.84–1.68), but also not significantly.

## Community practices and brucellosis transmission risk to humans

Not belonging to the Wamaasai ethnic group decreased the odds of using risky practices significantly, as did being of Muslim or Christian faith compared to having traditional beliefs (Table 6). Age, sex, education and knowledge were not associated with the level of transmission risk practices performed.

The number of Christians (32%) and Muslims (12%) in the study was smaller compared to those who followed traditional religious beliefs (56%). Results from the FGDs showed that overall culture played a bigger role in determining behavior than did religion or education alone. This is especially because these practices are a way of life and deeply entrenched and also due to the low perception of risk related to these practices.

## Attitudes related to the consumption of raw animal products and livestock handling

The results of this study show that the majority of the respondents engaged in risky behavior for human brucellosis transmission, as shown earlier in Table 2. During FGDs these practices were attributed to culture and long-term engagement in these practices without any perceived negative consequences. Participants in FGDs noted that long standing tradition, perceived benefits of raw animal product consumption, low risk perception attributed to these practices

**Table 6. Predictors for the use of practices by agro-pastoralists risking brucellosis infection for a) null model and b) final model.**

|  | Variable | Odds ratio | 95% CI | P-value |
|---|---|---|---|---|
| a) | Intercept | 1.47 | 1.07–2.01 | 0.01 |
|  | Sex | 0.98 | 0.84–1.16 | 0.89 |
|  | Age | 0.94 | 0.9–1.02 | 0.12 |
|  | Education | 0.8 | 0.68–0.94 | 0.01 |
|  | Religion (Christian) | 0.82 | 0.68–0.98 | 0.03 |
|  | Religion (Islam) | 0.48 | 0.36–0.64 | <0.001 |
|  | Ethnicity (Wasukuma) | 0.87 | 0.71–0.01 | 0.21 |
|  | Ethnicity (Other) | 0.32 | 0.2–0.42 | <0.001 |
|  | Knowledge | 1.01 | 0.99–1.01 | 0.05 |
|  | **Variable** | **Odds ratio** | **95% CI** | **P-value** |
| b) | Intercept | 1.10 | 0.98–1.23 | 0.08 |
|  | Education | 0.82 | 0.71–0.96 | 0.02 |
|  | Religion (Christian) | 0.84 | 0.71–1.01 | 0.06 |
|  | Religion (Islam) | 0.51 | 0.39–0.67 | <0.0001 |

and pragmatic reasons, as the main drivers for risk behaviors. Raw milk was perceived to have better nutritional value, more desirable taste and smell, increased satiation and reduce the potential harm from ingested toxic substances. The excerpts below elucidate this:

> *"Raw milk is more nutritious and causes our babies to gain weight fast and become big".*

Women FGD2, Kilombero

> *"When we drink raw milk we can go for up to two days without feeling hungry as the milk has a lot of important nutrients and makes us very satisfied. But once you boil the milk it becomes just like water and has little value".*

Men FGD2, Kilombero

Raw milk was also preferred as a cure for stomach ulcers and a source of hydration in the absence of drinking water. The participants did not attribute any diseases to raw milk consumption, instead associating diseases with supernatural factors. In one FGD men observed that:

> *"We know people who have TB and yet they boil their milk before drinking it. These diseases are from God and not from milk".*

Male FGD1, Kilombero.

Members from the Maasai community also reported to engage in the consumption of raw blood and raw meat for cultural reasons and as a source of replenishing lost blood when anemic and after child birth.

> *"We, the Maasai, enjoy raw blood and every time an animal is slaughtered we drink the blood. It is part of our culture. Whenever a woman gives birth, a sheep is slaughtered and before the woman eats anything else she is given blood to drink so that she can quickly regain her strength and also to replace all the blood she has lost".*

Women FGD3, Kilombero

Assisted parturition with bare hands was very common, especially among herdsmen, with the perception that as long as one washed their hands after, no harm was to be expected. Instead, any cases of fever in humans were associated with malaria. In one of the FGDs, men observed that they had heard from livestock officers that it was important to wear gloves when assisting in parturition. However, they indicated that, *"only the livestock officers use the gloves, we do not because it is our custom to use bare hands even if we have been told that we should wear gloves"*. This is further demonstrated in the excerpt below:

*"We have never encountered any harm after assisting in parturition either to the animal or to ourselves. So according to us, there can be no harm because if there were, we would have experienced it by now"*.

Men FGD1, Kilombero

More than half (67%) of the respondents fed aborted material to dogs while 28% and 4% buried it or left it to rot on the ground respectively. This task of disposal was performed by boys aged 10 to 12 years old. This data was corroborated by reports from FGDs where participants commented that they generally fed aborted material to dogs. They often cut and cooked the aborted material to prevent dogs from getting used to the taste and eating live calves. In an FGD with herdsmen they observed that:

"*Most often we feed the aborted material to our dogs. We cut it so as to drain most of the blood and cook the fetus before feeding it to the dogs so that dogs do not get into the habit of eating live calves"*.

Male FGD4, Kilombero

A small minority (3.3%) of the respondents reported to reside with livestock. This was not a common practice and most livestock keepers built an enclosure for animals outside. However, herd boys reported to sleep close to livestock to prevent attacks from predators and snakes, when living for many months in the wild, looking after cattle. This is shown through the following quote:

*"We sleep close to the cattle to avoid animals like snakes and lions. When we are close to cattle, if a wild animal comes the cattle are restless and we get startled and are not caught unawares"*.

Men FGD4, Kilombero

## Perceptions on wildlife-livestock interaction

Close to half (42%) of all the respondents in the structured interviews grazed their livestock in areas with wildlife. FGD participants observed that their livestock, especially cattle, routinely encountered wild animals such as puku antelopes, buffaloes, wild pigs, hippos, lions and even elephants. These encounters were due to the structure of land use in the villages whereby the grazing area directly neighbors a wildlife conservation area. This association was not considered as a potential route for transmission of disease by most of the respondents. Men in one FGD noted that,

*"These animals, especially buffaloes and antelopes, graze freely with our cattle. These do not cause any harm to cattle because they belong to the same family. If you observe keenly antelopes have the same kind of hooves as goats and buffaloes as cattle".*

Men FGD3, Kilombero

## Discussion

The success of public health interventions against brucellosis relies on prior information about any misconceptions, traditional beliefs and local practices of the targeted communities [18, 26]. This study aimed at ascertaining the discrepancies between risk awareness and perception regarding brucellosis by the agro-pastoralist communities in the Kilombero Valley. Our main findings show a lack of knowledge of transmission dynamics and symptoms of brucellosis. The risk perception for contracting human brucellosis due to the consumption of raw animal products and the handling of animals is low. Behaviors exposing the people to a high risk of zoonotic transmission are frequently practiced, including raw animal products consumption and assisting livestock births. Factors associated with having no knowledge of the disease are gender and level of education, with females and participants with little education more likely to have never heard of brucellosis. Our findings show that older agro-pastoralists are more likely to have a higher level of knowledge on brucellosis. Engaging in practices risking transmission was positively associated with belonging to the Wamaasai tribe. Agro-pastoralists of Muslim and Christian faith were less likely to engage in risky practices than people with traditional beliefs. Even though a high priority disease in Tanzania, with an incidence rate of as high as 28.2% in humans in some areas [26] and affecting livestock productivity, the agro-pastoralist communities in the Kilombero Valley are inadvertently unaware of the problem.

### Inadequate awareness and knowledge on brucellosis in livestock and humans

The study found very low awareness of brucellosis among agro-pastoralists and a lack of recognition of the disease as a public health threat. This presents a challenge to correctly identifying brucellosis symptoms and to actively addressing the disease. This is in line with other studies from various parts of Tanzania and Africa, which found that people had inadequate knowledge on the causes, signs and transmission pathways for brucellosis, especially among livestock keepers [13, 27–32]. One recent major review of studies on knowledge, attitudes and practices of brucellosis in Africa and Asia concluded, that, in line with our findings, about 50% of participants did not know about brucellosis [33]. This shows a key gap in knowledge, especially in Sub-Saharan Africa, where brucellosis has been determined to be endemic and a high priority disease [7]. A proper understanding of the lay knowledge, attitudes and behavioral practices risking transmission as well as perception regarding a disease is important for the development of suitable disease control strategies including community education and sensitization [6]. In order for control efforts, including livestock vaccination, to be effective, the engagement of livestock keepers is paramount. Prevention via vaccination needs to be accompanied by the adoption of less hazardous behavior, such as the proper disposal of aborted material and placentas and controlled animal movement [18]. These measures are seriously hampered due to inadequate knowledge on how brucellosis is transmitted between animals and to humans.

In our study, agro-pastoralists noticed brucellosis symptoms in their livestock, such as abortion, still birth and infertility, but attributed them to trypanosomiasis, unknown disease and supernatural factors. When attributed to the supernatural, they were related to similar

occurrences in human beings especially in the case of still births and not conceiving and thus accepted as part of life. More importantly, our study shows that participants did not consider there to be a problem, since possible brucellosis symptoms were witnessed infrequently, and since they owned large herds which was highly regarded as a status symbol. Owning a large herd was considered more important than ensuring that all the animals in the herd were productive. Therefore, infertility and low milk yield of some animals was not considered as a threat to the overall productivity allowing them to keep less productive and chronically sick animals. These sick animals have other values attributed to them, such as infertile cows being valued for their size or cows with low milk yield being thought of as calmer. Consequently, as a result of these attitudes and behavior, the potential for the spread of brucellosis in the herd is high. Furthermore, the study shows that the community felt capable of addressing brucellosis signs in livestock through conventional and traditional methods, such as using herbs and salt. In a study conducted in Cameroon, it was also shown that pastoralists did not remove chronically sick animals from their herds since they would fetch low prices in the market [34]. This shows that while livestock keepers often observe symptoms of disease in livestock, they have different reasons for not culling sick animals. This kind of behavior can perpetuate the spread of brucellosis to livestock and people and also threaten disease control efforts that involve culling of livestock.

## Culturally acceptable behavior risking transmission of brucellosis

A recent study conducted in Kenya linked brucellosis in humans to the consumption of unpasteurized milk [8]. However, results from the current study showed that consumption of raw animal products and close contact with animals and their products were not perceived as risk factors for disease. This relates to a long-standing tradition and cultural practice and engaging in practices risking transmission was linked to ethnicity in our study. Raw milk was reported to be regularly used as a replacement for drinking water. Lay perceptions about disease etiology have been identified to play a big role in determining whether people engage in risky behavior or not [35, 36]. It has been shown that humans can get infected with brucellosis through drinking raw milk, raw blood, residing with livestock and handling birth products without gloves [9, 37]. Comparably, in other studies, participants reported to engage in raw milk consumption, handling animal birth material and assisting in parturition with bare hands in other countries [8, 38–43]. Our study also shows that while some participants were informed of the need to avoid consumption of raw animal products and handling birth materials with bare hands, this did not lead to safer behavior. The main reasons cited for this were the cultural attachment they have to livestock and their products and the perceived low risk for disease transmission due to this behavior. In this study, age significantly increased the odds to know more about brucellosis, while being male and having better education were also correlated with increased knowledge but not significantly. Older people in agro-pastoralist settings have more experience in livestock management and consequently enhanced knowledge. Men too, have a greater role in treating sick livestock and thus have access to insights on livestock diseases which could explain why they are more likely to have better knowledge on brucellosis. These findings correspond to other studies which have shown a variable effect of age, gender and formal education on knowledge of brucellosis [44–47]. Here it was also found that adherence to culture was a stronger driver for engaging in behavior risking transmission, than formal education. Similarly, study participants in other settings engaged in the consumption of unpasteurized milk and unprotected assisted parturition in spite of increased knowledge on brucellosis [45, 48]. Pastoralists in Tanzania and Kenya, including the Wamaasai, were found to consider the adherence to the cultural practice of raw milk and blood consumption to be

more important than the protection from zoonotic disease [19, 36]. This demonstrates that increased knowledge on brucellosis in agro-pastoral systems did not correspond to a change in behavior, most likely due to the lack of perception of any problem. Importantly, it shows that sensitization on brucellosis risk factors and transmission dynamics is not enough. A greater level of engagement, considering the local socio-cultural context is necessary. For example, findings in our study show that the role of religion in affecting food consumption and husbandry behavior risking transmission needs to be explored further. Studies have shown that risk communication strategies must consider the social cultural background of the communities involved [49, 50]. This is because values, norms and the characteristics of the risk itself which include how prevalent the risk is, severity of the risk and the perceived effects of the risk to individuals and community, determine how risk is assessed and perceived [49, 51]. This determines what diseases and conditions are thus considered a priority for control by communities and affect their willingness to change their behavior.

In this study, men engaged in most of the livestock related duties such as slaughtering, herding and assisting in parturition. According to participants in this study, younger men (13–19 years old) were involved in preparation of aborted fetuses to feed to dogs. They also reported to heavily consume raw milk, not just as food, but to quench their thirst while herding. Consequently, this increases their exposure to brucellosis. This links in with prevalence studies conducted in Tanzania and Kenya, where men were reported to have a higher seroprevalence for brucellosis than women [52, 53]. The differential role that age and gender play in transmission and infection with zoonotic disease is well established and is attributed to the different roles in livestock keeping [19, 36, 54, 55]. The close interaction between humans, livestock and wildlife has been implicated in the perpetual transmission of brucellosis across species [15, 56]. This is through contaminated material such as birth products in the grazing areas and watering points as well as aerosol transmission. In this study, participants routinely interacted with wildlife including buffaloes, antelopes and hippos, but did not consider this as a potential pathway for brucellosis disease transmission to livestock or humans. Brucellosis transmission between wildlife and livestock is poorly understood and was only associated with tse tse flies by the agro-pastoralists. Our study found that behavior might differ based on religion, but ethnicity played a bigger role in perpetuating risk behavior.

This study adds insights into the awareness, risk perception, misconceptions and risk behaviors associated with brucellosis, and some of the reasons governing these attitudes and practices. Interestingly, and most alarming, is the evidence that agro-pastoralists in the Kilombero district do not perceive the disease symptoms they witness in livestock and people as a problem. Other studies have commented on the need to understand the reasons governing people's behavior through detailed qualitative studies [57]. It was suggested that this would help to develop better, contextually relevant health education and disease control plans. This is especially important because better knowledge on brucellosis does not always correspond with engaging in less risky practices [45, 48]. In addition, the perceptions of livestock keepers on the causes and symptoms of brucellosis and the risks associated with raw animal sourced foods and livestock handling practices play a big role in determining their behavior. Consequently, community education and sensitization that would be impactful and lead to long term behavior change would need to be very engaging and consistent so that the community would ask questions and have their concerns on risk perception addressed and responded to adequately.

## Conclusion

Our study has found a mismatch between real transmission risk and its perception by the communities, making risk communication as part of control efforts very difficult. Brucellosis is

largely unknown and its symptoms are not considered a big problem by the livestock keepers in the Kilombero district, Tanzania. Thus, brucellosis symptoms, while often observed, were not perceived as severe and a threat to people and overall livestock productivity. Knowledge on the signs, causes and risk factors for the disease in livestock and humans is inadequate, but even increased knowledge on brucellosis and formal education did not correspond to safer habits. How risk is perceived by health experts and lay people is often different and this has an impact on public health messaging and community sensitization [50]. In this study, the disparity in risk perception between professionals and the community is notable because the lay people did not consider animal handling and consumption of animal sourced foods as risk factors for brucellosis. Policy decisions therefore, need to include lay risk perceptions so as to develop targeted risk communication messages and to not only inform the community but engage them to deal with the low awareness and misconceptions. Control efforts too, such as surveillance, vaccination and safer animal handling behavior require the involvement of the agro pastoralists in order to be successful. This study recommends focused community engagement and sensitization to address the low awareness and misconceptions among agro-pastoralists.

## Limitations of the study

This study included the collection of qualitative data which means that the results of this study cannot be generalized. However, it provides in-depth information that can be used in aiding community sensitization and engagement in this context. The information generated through the structured interviews was self-reported on personal behavior which could result in recall bias and respondents giving answers that they might consider desirable. To mitigate this, triangulation, using multiple methods including observation ensured that the data was accurate and valid. Lastly, focus group discussions were conducted jointly with all the ethnic groups which could have resulted in a lack of clarity of ethnic differences.

## Supporting information

**S1 Strobe checklist.**
(PDF)

**S1 Quantitative data.**
(CSV)

**S1 Structured Interview questionnaire.**
(DOCX)

**S1 Qualitative tool. Focus group discussion guide.**
(PDF)

**S1 Qualitative data. Focus group discussions data.**
(PDF)

## Acknowledgments

We acknowledge all the participants in the study in all the three villages as well as the district, ward and village authorities for permission to conduct the study. Ifakara Health Institute is acknowledged for hosting the researcher during the data collection period. John Gitahi is also acknowledged for his help with initial quantitative data analysis.

## Author Contributions

**Conceptualization:** Caroline M. Mburu, Salome A. Bukachi, Kathrin H. Tokpa, Gilbert Fokou, Khamati Shilabukha, Bassirou Bonfoh, Rudovick Kazwala.

**Data curation:** Caroline M. Mburu, Salome A. Bukachi, Katharina Kreppel.

**Formal analysis:** Caroline M. Mburu, Salome A. Bukachi, Kathrin H. Tokpa, Mangi Ezekiel, Katharina Kreppel.

**Funding acquisition:** Kathrin H. Tokpa, Gilbert Fokou, Bassirou Bonfoh, Rudovick Kazwala.

**Investigation:** Caroline M. Mburu, Salome A. Bukachi.

**Methodology:** Caroline M. Mburu, Salome A. Bukachi, Kathrin H. Tokpa, Gilbert Fokou, Khamati Shilabukha, Mangi Ezekiel, Bassirou Bonfoh, Rudovick Kazwala.

**Project administration:** Kathrin H. Tokpa, Bassirou Bonfoh, Rudovick Kazwala, Katharina Kreppel.

**Resources:** Salome A. Bukachi, Kathrin H. Tokpa, Gilbert Fokou, Bassirou Bonfoh, Rudovick Kazwala.

**Software:** Caroline M. Mburu, Katharina Kreppel.

**Supervision:** Salome A. Bukachi, Kathrin H. Tokpa, Gilbert Fokou, Khamati Shilabukha, Mangi Ezekiel, Bassirou Bonfoh, Rudovick Kazwala.

**Validation:** Salome A. Bukachi, Kathrin H. Tokpa, Gilbert Fokou, Khamati Shilabukha, Mangi Ezekiel, Bassirou Bonfoh, Rudovick Kazwala, Katharina Kreppel.

**Visualization:** Salome A. Bukachi, Kathrin H. Tokpa, Gilbert Fokou, Khamati Shilabukha, Mangi Ezekiel, Bassirou Bonfoh, Rudovick Kazwala, Katharina Kreppel.

**Writing – original draft:** Caroline M. Mburu.

**Writing – review & editing:** Caroline M. Mburu, Salome A. Bukachi, Kathrin H. Tokpa, Gilbert Fokou, Khamati Shilabukha, Mangi Ezekiel, Bassirou Bonfoh, Rudovick Kazwala, Katharina Kreppel.

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
