## [Decision Letter · Decision Letter 0]

8 Sep 2020

Dear Dr. Mwihaki Mburu,

Thank you very much for submitting your manuscript "Lay attitudes and misconceptions and their implications for the control of brucellosis in an agro-pastoral community in Kilombero district, Tanzania" for consideration at PLOS Neglected Tropical Diseases. As with all papers reviewed by the journal, your manuscript was reviewed by members of the editorial board and by several independent reviewers. Your manuscript was reviewed by three experts in the field who are knowledgeable on the geographical region where the study was conducted and its socio economical characteristics. While the three of them acknowledge the relevance of the scientific question and understand that this paper provides useful information within a one health approach, they all agree on some limitations of the study that must be addressed before considering its publication in the journal. In particular, they found weaknesses in statistical analysis, expression of the results and language quality. They suggest to improve the paper by enhancing the linkage between quantitative and qualitative data among a series of major and minor recommendations included in their revisions. In light of the reviews (below this email), we would like to invite the resubmission of a significantly-revised version that takes into account the reviewers' comments. 

We cannot make any decision about publication until we have seen the revised manuscript and your response to the reviewers' comments. Your revised manuscript is also likely to be sent to reviewers for further evaluation.

Sincerely,

Esteban Chaves-Olarte

Guest Editor

Hélène Carabin

Deputy Editor

Reviewer's Responses to Questions

**Key Review Criteria Required for Acceptance?**

**Methods**

-Are the objectives of the study clearly articulated with a clear testable hypothesis stated?

-Is the study design appropriate to address the stated objectives?

-Is the population clearly described and appropriate for the hypothesis being tested?

-Is the sample size sufficient to ensure adequate power to address the hypothesis being tested?

-Were correct statistical analysis used to support conclusions?

-Are there concerns about ethical or regulatory requirements being met?

Reviewer #1: Generally, lme4 is used for generalized linear mixed effect models setting random effects. However, the analysis was performed by glm. Explain why lme4 was used.

P7 line7

I think binomial distribution should be used instead of Poisson distribution.

P10 line 17

“over 50 years of age” This result just shows the analysis of comparison between age category 18-25 and all other age categories. If the authors’ interest is to compare the youngest category and other categories, keep current form of analysis but revise the sentence, taking my comment into account. If the authors hypothesis is that knowledge or practice improve as people get older, change the age categories into rank (for example 18-25:1, 26-33:2, 34-41:3, …) and analyze it by glm. The same goes for education.

Reviewer #2: This papers addresses an important area of focus, namely, the interface between livestock and humans, especially for pastoral/agro-pastoral communities especially across Africa. In Kenya and Tanzania this is especially important for the Maasai due to there close association with livestock. In addition, the Maasai also inhabit areas which bring them into considerable interaction with wildlife. The study explores the disparities in risk perception and the practices that raise the risk of transmission in an agro-pastoral setting. In my view, this objective is clearly stated.

The study design is appropriate and the study population clearly described. 

It is not clear what the sample size is. On page 5 line 20 the authors indicate that 333 structured interviews and 7 focus group discussions were done. However, on the same page 5 line 25, the authors indicate that all 352 agro-pastoralist households were visited. This requires clarification. 

Some of the aspects/data collected for this work, e.g. wildlife-livestock-human interface (page 6 line 9) has not come out in the data presented and, hence, in the discussion.

The authors indicate (page 7 line 21) that "For practices, the respondent reporting the riskiest practices scored the highest." What about the scores for knowledge? Was knowledge also similarly ranked?

Reviewer #3: - The objectives are clear

- The design is well articulated 

- The population is appropriate

- Sample size is okay

- The analyses need revistation

**Results**

-Does the analysis presented match the analysis plan?

-Are the results clearly and completely presented?

-Are the figures (Tables, Images) of sufficient quality for clarity?

Reviewer #1: Demographic characteristics

Show all the data by table or figure. The information about age and ethnic group are unclear. In terms of ethnic group, quite small group, less than 5 people for example, can be grouped as other group, but otherwise show all the data.

P8 line 12-13

Traditional religion (56%) is inconsistent with the data in EXCEL file. Carefully check this contradiction.

“Most of them identified-----” Do not use “Most” for 56%.

P8 line 19-20 “their causes were not associated with brucellosis as shown in Table 2”

Current expression means causality between brucellosis symptoms and its cause was cleary identified by robust, scientific methods. But this is based on questionnaire survey so the ture cause is not identified. Rephrase it.

Does Table 2 mean Fig 2?

Table 1

The readers have an impression that tables of awareness and causes have some relations since they are expressed in parallel. Rearrange table.

In the table for awareness, the relationship between question and frequency is a bit unclear. Revise it.

P10 line 2-3

It’s not clear for me to show the data from just the 24 respondents who had heard of brucellosis. If the authors want to show the major brucellosis symptoms recognized by farmers, the data from all farmers should be used.

Fig 2

In title, it’s unclear what “minority” means. Revise the title.

“Percentage” should come to left side.

P10 line 33-P11 lin2

The method of analysis is not described.

Table 2

Show the category to compare in Explanatory variables. Except for sex, the category compared is not clear.

In CI, it’s not clear whether “-” shows minus or range of CI. It seems there are mistakes. Revise it.

Practice-Education (secondary): Coefficient or CI is wrongly expressed.

Table 3 is not found.

Table 4

I do not think this is from the final model since all the explanatory variables are included.

P16 line10

What are the sources of the data?

P17 line31 

“since they owned large herds” The logic is unclear. Clearly explain it.

P18 line26

“Positive” is unsuitable word here. Rephrase the sentence.

P18 line27-29

The logic is unclear. Please clarify it.

P19 line15-20

Describe the information about age in the references.

Reviewer #2: The authors present the results starting first with quantitative and followed with qualitative data. The captions for the Tables and Figures should be more succinct. It is also important to include the N in the Tables in order to put the percentages into perspective. 

Table 1 (page 8) is difficult to follow. For instance, for have you heard of brucellosis in livestock the frequency is 7.2% (it would be nice to indicate the actual count as well). Now, for the perceived causes, which is given next, do those percentages derive from the 7.2%(n??) respondents? One is unable to make this decision from the Table. 

The caption for Table 2 is a too long (it is, in fact, two sentences long). The table also includes under practice Ethnicgroup1. What is the justification of including this as a practice? 

Where is Table 3 in the paper? 

The caption for Table 4 also should also be made more succinct. Also for this Table, I am looking for an explanation of the link between education and religion in terms of their relationship with practices. Could we get some additional information on this, especially for the qualitative data? 

I fail to get the rich narrative that should come from the qualitative data. At best there are many short quotations but the authors seem to have failed in getting the story out in a clear and concise way supported by the quotations. Reading through the qualitative section, one is not sure whether their was a more general concurrence from the 7 FGDs and, which is represented by the quote or not. The authors should review this section. 

On page 16 line 12: I am still trying to figure out how cattle interact with ... hippos and lions ... How is the term interaction used in this sense? 

The caption for Figure 2 should be reviewed and include the percentages to provide a little more information.

Reviewer #3: - The qualitative part will need improving

**Conclusions**

-Are the conclusions supported by the data presented?

-Are the limitations of analysis clearly described?

-Do the authors discuss how these data can be helpful to advance our understanding of the topic under study?

-Is public health relevance addressed?

Reviewer #1: Describe the limitations of the study.

Reviewer #2: What are the limitations, if any, for this study? This study has significant implications on communication of public health messages. The authors should build a case based on the data presented. At present one cannot get the public health argument. On page 20 line 20 it is indicated that "This disparity in risk perception, thus, presents a challenge in risk communication related to animal handling and consumption of animal sourced foods." What disparity is being referred to here? In the previous sentence the authors write about the disparity in risk perception between the health experts and lay people. One may argue that the disparity is always there between the experts and lay people. How this creates a challenge in risk communication is not clear. In my view, a challenge in risk communication would arise if the lay people have two or more competing risk perceptions. This should be clarified.

Reviewer #3: - The conclusions are okay however there are many limitations and have not been well addressed

**Editorial and Data Presentation Modifications?**

Reviewer #1: P1 line 21-22

The number of FDGs is complicated.

P2 line 33

“questions and concerns” This should be described in detail in discussion.

Introduction

In the current form the readers have an impression that brucellosis is endemic just in sub-Saharan Africa but the disease is also endemic in other areas. Rephrase Introduction.

P3 line30

The reference 13 does not describe the information.

P3 line 34-P4 line4

Epidemiological study and implementation of disease control are largely different. Describe the examples of the implementation of disease control and the results.

The information is complicated among Study design, Sampling and Data collection. For example, if this is a census among the 3 villages, explain it before the number of interviews. Avoid repeating same information and reorganize these parts.

Describe the reason not to include some agro-pastoral households in structured interview.

Men and women were interviewed separately in FGDs. Make it clear whether the authors count this separated discussion as one or two.

Describe how many participants joined in a FDG.

P6 line34 - p7 line3 

Make it clear the explanatory variable and response variable.

Show all the 29 and 12 categories for knowledge and practice levels, respectively. Either in methods or results is fine.

Cleary describe univariable and multivariable analyses in method and result. It is not clear which result is shown in the table in result part.

P7 line21-22

The sentence is not clear for me. Rephrase it.

P7 line 33 

Attach the reference for “supported by existing literature”

Reviewer #2: The paper require editing through out. In particular, the Table and Figure captions should be reviewed. The titles could be made more succinct. I have made additional suggestions in the preceding sections.

Reviewer #3: (No Response)

**Summary and General Comments**

Reviewer #1: Although the data collected in the study is important, the current manuscript has weaknesses in statistical analysis, expression of the results and language quality.

I recommend to have advice for statistical analysis.

I also advise the authors work with a writing coach or copyeditor to improve the flow and readability of the text.

Reviewer #2: This paper provides useful information within a one health approach. The authors have attempted to make an argument on the interaction between wildlife-livestock-human interface but the argument seems lost in the weak link between quantitative and qualitative data. Authors could improve the paper by enhancing the linkage between quantitative and qualitative data.

Reviewer #3: The study aimed to assess perceptions understanding of brucellosis in an agro-pastoral community in Kilombero district, Tanzania. This is a critical study given the insufficient number of social and cultural data on zoonotic diseases. However, one main finding, a total lack of knowledge about brucellosis among the participants, baffles me. Perhaps this emanates from how the data collection was implemented, questions asked, in particular, the qualitative data, as I explain below.

Methodology

Having worked with Sukuma and Mangati pastoral groups bordering the Morogoro region, I find it rather odd to read that the interviews were conducted in Kiswahili rather than their mother tongue and without the use of an interpreter (Maasai, possibly, however, the Mangati?). I say these because most are not fluent in Kiswahili (Kindly use the native name Kiswahili), let alone conversing about livestock diseases using a “second” language i.e. Kiswahili. The authors should clearly state this limitation and explain how they think; a clear and meaningful understanding of lay perceptions on zoonotic diseases may have been lost in translation. In addition, it is not stated whether the FGDs were grouped according to the ethnic groups of the participants and not. Why was this not done? And about my observation above, if they were grouped together, how possible was the moderation given the diversity of the participants' indigenous language?

It would be interesting to know how exactly, even in Kiswahili, the idea of brucellosis was communicated to the participants during the discussions. If the guide only queried about zoonosis as a category of diseases that humans get from livestock, then a lot of lay information may have been missed. In the transcriptions attached, the interviewer asks the respondents if they have heard about brucellosis. What Kiswahili word was used? Does it even exist? Wouldn’t the answer obviously be negative? If participants knew brucellosis in animals, should we read that the same understanding “abortions”, was communicated to the participants and the results could be wrong? No wonder the results section on brucellosis perceptions is so short (line 28-34) but it carries the study.

Percentages have been used in the qualitative results section in a rather odd fashion, yet this was not clearly justified or referenced in the qualitative data analysis strategy section. I think they do not add any value given that there is a whole section on quantitative results already. If these are from the KAP survey results and were used to elaborate on the quali data, then the design would have to be re-written to conform to the principles of the Mixed methods approaches. A number of frameworks do exist to guide on the same (For example Creswell, J. W., & Clark, V. L. P. (2017). Designing and conducting mixed methods research (Third). Thousand Oaks, CA: Sage publications.)

The qualitative results could be improved through thoroughly unpacking comparatively notions of zoonotic diseases in the pastoral groups involved, given that the study clearly demarcated them in terms of their ethnic groups (Maasai, Wamangati and Sukuma). It would be interesting to the readers to know how each ethnic group discussed brucellosis instead of lumping lack of understanding across all the ethnic groups. I say this because brucellosis tests are now widely available in many district hospitals, and as one of the study quoted found that the knowledge of some zoonotic diseases among the Maasai, for instance, is increasing because of these health facilities visitations and upon receiving diagnostic results. For example, as a result, some pastoralists and agro pastoralists in some areas of Tanzania now widely speak of “brusela” (localized word) as a “new” disease, emanating from the consumption of livestock products such as milk. To improve the paper, perhaps it will be useful to obtain information from the surrounding health facilities whether such tests are available or have been performed.

PLOS authors have the option to publish the peer review history of their article (what does this mean?). If published, this will include your full peer review and any attached files.

Reviewer #1: Yes: Asakura Shingo

Reviewer #2: No

Reviewer #3: No
---

## [Decision Letter · Decision Letter 1]

4 Jan 2021

Dear Dr. Mwihaki Mburu,

Thank you very much for submitting your manuscript "Lay attitudes and misconceptions and their implications for the control of brucellosis in an agro-pastoral community in Kilombero district, Tanzania" for consideration at PLOS Neglected Tropical Diseases. As with all papers reviewed by the journal, your manuscript was reviewed by members of the editorial board and by several independent reviewers. The reviewers appreciated the attention to an important topic. Based on the reviews, we are likely to accept this manuscript for publication, providing that you modify the manuscript according to the review recommendations. 

Sincerely,

Esteban Chaves-Olarte

Guest Editor

Hélène Carabin

Deputy Editor

Reviewer's Responses to Questions

**Key Review Criteria Required for Acceptance?**

**Methods**

-Are the objectives of the study clearly articulated with a clear testable hypothesis stated?

-Is the study design appropriate to address the stated objectives?

-Is the population clearly described and appropriate for the hypothesis being tested?

-Is the sample size sufficient to ensure adequate power to address the hypothesis being tested?

-Were correct statistical analysis used to support conclusions?

-Are there concerns about ethical or regulatory requirements being met?

Reviewer #1: P6 line7

This should be moved to Result part.

In terms of ethnic group, quite small group, less than 5 people for example, can be grouped as other group, but otherwise show all the data. Especially Wamaasai should be separately listed in demographic characteristics (Table 1).

Data management and analysis

Explain statistical analysis in more detail referring the comments below and avoid redundancy through this part.

Education and age group have more than 2 categories. Explain which data type was used for these explanatory variables in analysis; for example, nominal scale or ordinal scale.

Show the 29 and 12 categorical variables for knowledge and practice score and the simple results of them before statistical analysis.

Reviewer #2: My earlier comments have been addressed

Reviewer #3: (No Response)

**Results**

-Does the analysis presented match the analysis plan?

-Are the results clearly and completely presented?

-Are the figures (Tables, Images) of sufficient quality for clarity?

Reviewer #1: Table 2

The answers of categories 2-5 are affected by the existence and the frequency of the symptoms in cattle herd. It is not suitable to show the knowledge of the disease name and observation of the symptoms in a same table with the current title.

Fig 2

It is not clear what the Fig shows due to inaccurate title. Moreover, integrate how to express the result with other data if the Fig is necessary.

P10 line4-6

It is not clear what the sentence indicates due to the language quality. In addition, why only this result is expressed by text?

P10 line 10

“They were not linked to brucellosis” Revise the sentence.

Table 3

It seems the respondents were asked to choose one category of the cause of each symptom because total % is 100 in each symptom. Questionnaire interview was incorrect because it excluded the possibility that the respondents knew several causes of the symptom. 

Table 4 

Relative risk is only used for cohort study or randomized control trial. Explain why relative risk is used.

Tables 4 and 5

Show all the result including non-significant factors.

Cleary explain whether the result is from univariable or multivariable analysis.

Refer other papers to learn how to express the result of statistical analysis.

Reviewer #2: Earlier comments addressed

Reviewer #3: (No Response)

**Conclusions**

-Are the conclusions supported by the data presented?

-Are the limitations of analysis clearly described?

-Do the authors discuss how these data can be helpful to advance our understanding of the topic under study?

-Is public health relevance addressed?

Reviewer #1: (No Response)

Reviewer #2: Earlier concerns addressed

Reviewer #3: (No Response)

**Editorial and Data Presentation Modifications?**

Reviewer #1: Abstract 

P1 line 27-28

Here is inconsistent with the result.

Introduction

P3 line19

The information about Ref 15 is not enough.

Reviewer #2: (No Response)

Reviewer #3: (No Response)

**Summary and General Comments**

Reviewer #1: The manuscript still has a room for improvement in statistical analysis, expression of the results and language quality.

Reviewer #2: The authors have made a good attempt to responD to issues I raised in the first review. However, in my initial review I also raised a fundamental issue touching on authorship. The authors have for reasons unknown ignored to respond to this issue. In my earlier review, I asked what role Dr. Khamati Shilabuka played as an author. While the roles played by each author, other than Dr. Shilabuka, have been identified, there is collective silence on the issue of Dr. Shilabuka. In fairness to Dr. Shilabuka, if there was no contribution then the name should be removed. Otherwise the authors should respond to this concern.

Reviewer #3: (No Response)

PLOS authors have the option to publish the peer review history of their article (what does this mean?). If published, this will include your full peer review and any attached files.

Reviewer #1: Yes: Asakura Shingo

Reviewer #2: No

Reviewer #3: No
---

## [Decision Letter · Decision Letter 2]

6 Apr 2021

Dear Dr. Mburu,

Thank you very much for submitting your manuscript "Lay attitudes and misconceptions and their implications for the control of brucellosis in an agro-pastoral community in Kilombero district, Tanzania" for consideration at PLOS Neglected Tropical Diseases. As with all papers reviewed by the journal, your manuscript was reviewed by members of the editorial board and by several independent reviewers. The reviewers appreciated the attention to an important topic. Based on the reviews, we are likely to accept this manuscript for publication, providing that you modify the manuscript according to the review recommendations. As you can see, reviewer 1 still has several issues that need to be accurately addressed before being able to accept this work. Please pay close attention to all the minor issues raised and described by the reviewer and modify the manuscript accordingly. More importantly, for the results presented in table 4, and considering that the questionnaires were applied without giving the participants the opportunity to select more than one option, it is crucial that you acknowledge the limitations of the study on that respect in the discussion. In addition, indicate how your conclusions are still valid even if these limitations exist. 

Sincerely,

Esteban Chaves-Olarte

Guest Editor

Hélène Carabin

Deputy Editor

Reviewer's Responses to Questions

**Key Review Criteria Required for Acceptance?**

**Methods**

-Are the objectives of the study clearly articulated with a clear testable hypothesis stated?

-Is the study design appropriate to address the stated objectives?

-Is the population clearly described and appropriate for the hypothesis being tested?

-Is the sample size sufficient to ensure adequate power to address the hypothesis being tested?

-Were correct statistical analysis used to support conclusions?

-Are there concerns about ethical or regulatory requirements being met?

Reviewer #1: Page1 line 22

“gender” is vague. Revise the sentence to clarify which sex has positive or negative association.

Page1 line24-25 

“Following traditional religion and belonging to the Wamaasai ethnicity increased the odds of carrying out these practices.”

It is not clear whether the result was significant. If not, delete it.

Page1 line28-30 

“7.2%” is only for the 1st item, not for 2nd and 3rd items within the sentence. Revise it.

P6 line24-26

It is not clear which model was used for what association. Cleary explain it. Moreover, if ZINB model was used for the association between socio-demographic variables and practices, it is wrong.

P6 line29-30

Describe how education and age were changed into ordinal scale.

P7 line14-25

This should be expressed before the explanation of analysis and models. Data management and analysis part is not organized well. Put the information in order.

P7 line16

Remove “s”. This does not appear anywhere else. 

P7 line22-25

Delete. The sentence should be in Result and there is.

Reviewer #2: None. My earlier comments had been addressed in the previous revision.

**Results**

-Does the analysis presented match the analysis plan?

-Are the results clearly and completely presented?

-Are the figures (Tables, Images) of sufficient quality for clarity?

Reviewer #1: Tables 1-4

Integrate the format throughout the tables.

P9 line5-7

This should be in Discussion, not in Result.

P9 line7, Table 2

P11 line2-3, line11-12

Here quantitative results should be mentioned. No need to describe qualitative comment.

Table 3 

It is not clear which answer was returned. The result of second category is incorrect.

Table 2

This is not well organized. Current table seems to be original format of the questionnaire which is not suitable for readers. Organize it to be easily understood.

Table 4

This is my previous comments to the Table.

“It seems the respondents were asked to choose one category of the cause of each symptom because total % is 100 in each symptom. Questionnaire interview was incorrect because it excluded the possibility that the respondents knew several causes of the symptom. “

The authors’ answer does not make sense, and the association with the result of the table and qualitative data is not the issue here. If the authors want to use the result, limitation of the interview and thus the result should be clearly mentioned. The readers do not think the result is from open question, and it should have been even possible for the interviewees to answer several causes of symptoms in open question, which was not considered in the survey.

Judging from the authors’ comment, not only this result but all results are now doubtful. Precise explanation about the questionnaire items and the structure of the questionnaire (open/closed question for all the items) must be added in Material and methods, and the way of the expression of all quantitative results should be checked again through the manuscript.

Tables 5 and 6

Since the table shows the result of multivariable analysis, show the result of final model which includes only significant factors. In that case, the result of two models should be expressed separately because they must have different factors. Be careful that the Odds ratio, 95% CI and p-value will be different between the current table and final model. Since this research did not perform univariable analysis, the factors included in the analysis must be clearly described in Material and methods.

Reviewer #2: my earlier comments had been addressed.

**Conclusions**

-Are the conclusions supported by the data presented?

-Are the limitations of analysis clearly described?

-Do the authors discuss how these data can be helpful to advance our understanding of the topic under study?

-Is public health relevance addressed?

Reviewer #1: (No Response)

Reviewer #2: my earlier comments had been addressed.

**Editorial and Data Presentation Modifications?**

Reviewer #1: (No Response)

Reviewer #2: --

**Summary and General Comments**

Reviewer #1: (No Response)

Reviewer #2: In this revised version, the authors have responded to my concern. I am now satisfied.

PLOS authors have the option to publish the peer review history of their article (what does this mean?). If published, this will include your full peer review and any attached files.

Reviewer #1: Yes: Shingo Asakura

Reviewer #2: No

Figure Files:

Data Requirements:

Reproducibility:

References

---

## [Editor Report · Decision Letter 3]

21 May 2021

Dear Dr. Mburu,

We are pleased to inform you that your manuscript 'Lay attitudes and misconceptions and their implications for the control of brucellosis in an agro-pastoral community in Kilombero district, Tanzania' has been provisionally accepted for publication in PLOS Neglected Tropical Diseases.

Best regards,

Esteban Chaves-Olarte

Guest Editor

Hélène Carabin

Deputy Editor

---

## [Editor Report · Acceptance letter]

8 Jun 2021

Dear Mrs. Mburu,

We are delighted to inform you that your manuscript, "Lay attitudes and misconceptions and their implications for the control of brucellosis in an agro-pastoral community in Kilombero district, Tanzania," has been formally accepted for publication in PLOS Neglected Tropical Diseases.

Best regards,

Shaden Kamhawi

co-Editor-in-Chief

Paul Brindley

co-Editor-in-Chief
